# The human microbiome and COVID-19: A systematic review

**Shinya Yamamoto****, Makoto Saito, Azumi Tamura, Diki Prawisuda, Taketoshi Mizutani***, **Hiroshi Yotsuyanagi**

Division of Infectious Diseases, Advanced Clinical Research Center, Institute of Medical Science, The University of Tokyo, Tokyo, Japan

* miztanit@ims.u-tokyo.ac.jp

## Abstract

### Background

Human microbiotas are communities of microorganisms living in symbiosis with humans. They play an important role in the host immune response to respiratory viral infection. However, evidence on the human microbiome and coronavirus disease (COVID-19) relationship is insufficient. The aim of this systematic literature review was to evaluate existing evidence on the association between the microbiome and COVID-19 in humans and summarize these data in the pandemic era.

### Methods

We conducted a systematic literature review on the association between the microbiome and COVID-19 in humans by searching PubMed, Embase, and the Cochrane Library, CINAHL, and Web of Science databases for articles in English published up to October 31, 2020. The results were analyzed qualitatively. This study is registered with PROSPERO (CRD42020195982).

### Results

Of the 543 articles identified by searching databases, 16 in line with the research objectives were eligible for qualitative review: eight sampled the microbiome using stool, four using nasopharyngeal or throat swab, three using bronchoalveolar lavage fluid, and one using lung tissue. Fecal microbiome dysbiosis and increased opportunistic pathogens were reported in COVID-19 patients. Several studies suggested the dysbiosis in the lung microbiome of COVID-19 patients with an abundance of opportunistic pathogens using lower respiratory tract samples. The association between COVID-19 severity and the human microbiome remains uncertain.

### Conclusion

The human fecal and respiratory tract microbiome changed in COVID-19 patients with opportunistic pathogen abundance. Further research to elucidate the effect of alternation of the human microbiome in disease pathogenesis is warranted.

**Data Availability Statement:** All the extracted data are presented in Table 1 and Fig 1, and the used search terms are detailed in the supplement material 1 for ensuring reproducibility.

**Funding:** No funding was received for this study.

**Competing interests:** The authors declare no competing interest.

## Introduction

Microbiota widely colonizes the human body, and the human microbiota varies between individuals and ethnicities [1]. Although the role of the human microbiota has not been fully elucidated, the human microbiome is currently considered to be associated with several disorders, including inflammatory bowel disease [2], type 2 diabetes [3], Parkinson's disease [4], and colorectal cancer [5]. Among them, infectious diseases such as respiratory diseases are directly or indirectly associated with specific microorganism patterns. For example, the human upper respiratory tract microbiome in influenza patients is disturbed with *Pseudomonadales* abundance [6].

Coronavirus disease 2019 (COVID-19), caused by severe acute respiratory syndrome coronavirus 2 (SARS-CoV-2), was declared a pandemic by the World Health Organization on March 11, 2020. COVID-19 is a respiratory disease with a broad range of clinical manifestations, from asymptomatic or mild disease with cough and fever to severe pneumonia with multiple organ failure and acute respiratory disease syndrome (ARDS) [7]. ARDS is known to be caused by a cytokine cascade, including interleukin (IL), granulocyte colony-stimulating factor (G-CSF), and tumor necrosis factor (TNF). A previous study revealed that intensive care unit (ICU) patients with COVID-19, including ARDS, had an abundance of proinflammatory cytokines, including IL-2, IL-7, IL-10, GCSF, IP10, MCP1, MIp1A, and TNFα, compared to non-ICU patients [8]. These inflammatory cytokines were reportedly correlated with a specific pattern of the gut microbiome [9].

In the research of severe COVID-19, the interaction between the microbiome and cytokine cascade received considerable attention. However, the disparity factors of COVID-19 incidence and death among races, ethnicities, or countries remain uncertain at this moment. Hence, this systematic literature review aimed to analyze existing evidence on the association between the microbiome and COVID-19 in humans and summarize data on the microbiome in COVID-19 in the pandemic era. Moreover, in this review, the relationship between the human microbiome and COVID-19 severity has been investigated.

## Methods

All procedures used in this systematic review complied with the Preferred Reporting Items for Systematic Reviews and Meta-analysis guidelines. This study is registered to PROSPERO (CRD42020195982).

### Selection criteria and search strategy

This systematic review considered articles that were original clinical studies of any designs examining the human microbiome in COVID-19 patients. Reviews and editorial articles were manually checked for any additional relevant information. The MEDLINE/PubMed, EMBASE, Cochrane Database of Systematic Reviews, CINAHL, and Web of Science databases were systematically searched for published articles until October 31, 2020. The search terms used were as follows: ("severe acute respiratory syndrome coronavirus 2" OR "SARS-CoV-2" OR "SARS2" OR "Wuhan coronavirus" OR "coronavirus" OR "COVID-19" OR "novel coronavirus" OR "COVID19" OR "nCoV" OR "coronavirus disease 2019") AND ("microbiota*" OR "microbiome*" OR "mycobiome" OR "virome" OR "flora"). Two authors (S.Y. and A.T.) independently reviewed titles and abstracts of the identified studies written in English, and discrepant results were resolved via a second assessment. The search strategy and method used for selecting studies are summarized in S1 File. The risk of bias was assessed using the Risk of Bias Assessment tool for Non-randomized Studies (RoBANS) [10] for each study.

### Data synthesis

Data related to authors, study type, population, characteristics of patients and controls, microbiota community, microbiome characterization methods, sample collection time, key findings, treatments, and outcomes were extracted and recorded from all studies by the two authors independently. The results of this study were organized in a qualitative synthesis, including the relationship between microbiome and COVID-19 with available key study findings.

## Results

### Study characteristics

Our systematic search of the literature identified 543 articles for screening (Fig 1). Eighty-one full-text articles were examined for eligibility, and 16 studies that reported at least 296 patients were included in this review [11–26], including three unpublished articles, which were not peer-reviewed [21–23], and two articles [24, 25] with abstracts only (all written in English). All 16 articles were observational studies. Most of them were conducted in China (14/16 studies) [4–26], and all studies were on hospitalized adults. The microbiota community was sampled from stool (8/16) [11–14, 22, 24–26], nasopharyngeal or sputum or throat samples (4/16) [15, 16, 21, 23], bronchoalveolar lavage fluid (BALF; 3/16) [17, 19, 20], and lung tissue (1/16) [18]. No studies on skin, oral, vaginal, or urinary microbiota were reported. Eight studies [11–13, 16, 17, 19–21] used whole-genome sequencing to analyze the microbiome, and five studies [14, 15, 18, 23, 26] used 16S rRNA sequencing. Three articles (one unpublished article [22] and two abstracts [24, 25]) had no sequencing information. The findings of these studies are summarized in Table 1. The time point at sample collection

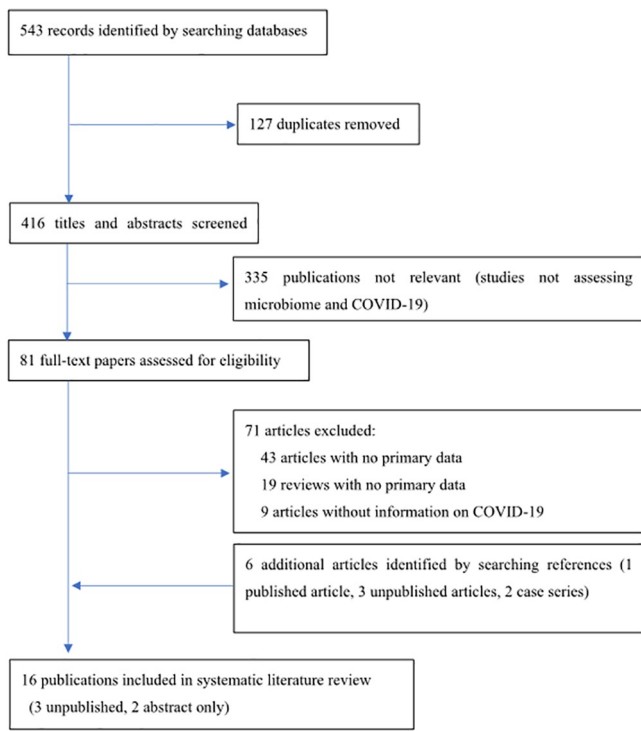

**Fig 1. PRISMA flow diagram.**

**Table 1. The characteristics of all included studies.**

| Study | Population and characteristics | Control | Samples or Microbiota community | Timepoint at sample collection | Methods of microbiome characterization | Change in microbiota | Treatment/ Outcome | Other key findings |
|---|---|---|---|---|---|---|---|---|
| Gu et al. (2020) | 30 COVID-19 patients, China / 13 were women / The median age was 55 (Range 29–70) years / 10 patients had comorbidities, including hypertension (30%) | (1) 24 H1N1 flu patients / (2) 30 healthy controls (matched for age, sex, body mass index) | Fecal samples | Samples were collected at admission. | V3-4 16S rRNA gene sequencing | Patients with COVID-19 or influenza had reduced bacterial diversity compared with controls. Stool samples of COVID-19 patients had an abundance of opportunistic pathogens, such as *Streptococcus*, *Rothia*, *Veillonella*, and *Actinomyces*. In contrast, *Ruminococcaceae* family and several genera from the family *Lachnospiraceae* were reduced in COVID-19 patients. | No outcome data | Richness, diversity, and structure of the gut microbiota were not significantly different between general COVID-19 and severe COVID-19. |
| Zuo et al. (2020a) | 15 COVID-19 patients, China / 7 were males, / The median age was 55 (22–71) years / 11 patients had moderate-severe COVID-19 / Comorbidities: Hypertension (4), Hyperlipidemia (4), Diabetes (2), etc. | 15 healthy controls / 6 community-acquired pneumonia | Stool samples | 2–3 times per week during hospitalization | Whole-genome sequencing | COVID-19 patients who were antibiotic naïve had increased opportunistic pathogens compared to controls. | Among 15 COVID-19 patients, 7 were antibiotic-naive, 8 received empiric antibiotics. | *Clostridium ramosum* and *C. hathewayi* were positively associated with severe COVID-19. In contrast, *Alistipes onderdonkii* and *Faecalibacterium prausnitzii* were negatively correlated with severe COVID-19. |
| Zuo et al. (2020b) | 30 COVID-19 patients, China / 16 were males, / The median age was 46 (15–71) years / 11 patients had comorbidities | 30 healthy controls / 9 community-acquired pneumonia | Stool samples (Fungal microbiome) | 2–3 times per week during hospitalization | Whole-genome sequencing | The mycobiome in most patients (22 of 30 COVID-19 patients) was similar to that in healthy controls. In contrast, gut microbiome in COVID-19 patients (8 of 30) had alterations, with enrichment of *Candida albicans* and heterogeneous composition. | No treatment data | COVID-19 patients had increased proportions of opportunistic pathogens (*Candida albicans*, *C. auris*, *Aspergillus flavus*) during hospitalization. |
| Zuo et al. (2020c) | Same population as Zuo et al. (2020a) | None | Stool samples | 2–3 times per week during hospitalization | Whole-genome sequencing | Stool with high SARS-CoV2 infectivity had a higher abundance of bacterial species, including *Collinsella aerofaciens*, *Collinsella tanakaei*, *Streptococcus infantis*, and *Morganella morganii*. | Same treatment as Zuo et al. (2020a) | Stool with low to no SARS-CoV-2 infectivity had higher abundances of *Parabacteroides merdae*, *Bacteroides stercoris*, *Alistipes onderdonkii*, and *Lachnospiraceae bacterium*. |
| Zhang et al. (2020) | 24 COVID-19 (nasopharyngeal swab) patients / 14 COVID-19 (sputum) patients, China / 37% were females / The median age was 40.5 (25–82) years | Pneumonia cases / 36 non-COVID-19 (nasopharyngeal swab) / 39 non-COVID-19 (sputum) | Nasopharyngeal swab, sputum samples | Samples were collected during inclusion in the cohort. | Whole-genome sequencing | COVID-19 patients had reduced alpha diversity in the airway microbiome. Opportunistic pathogens, including *Candida albicans* and human alphaherpesvirus 1, were frequently detected. | No treatment data | None |

*(Continued)*

**Table 1.** (Continued)

| Study | Population and characteristics | Control | Samples or Microbiota community | Timepoint at sample collection | Methods of microbiome characterization | Change in microbiota | Treatment/ Outcome | Other key findings |
|---|---|---|---|---|---|---|---|---|
| De Maio et al. (2020) | 18 COVID-19 patients, Italy<br><br>All patients had non-critical COVID-19 | 22 healthy controls | Nasopharyngeal swab samples | Samples were collected because of suspected COVID-19. | Sequencing of V5-V6 hypervariable region of bacterial 16S rRNA | The microbiota of the nasopharynx was not different in COVID-19 patients compared with that in controls. | No treatment data | None |
| Shen et al. (2020) | 8 COVID-19 pneumonia patients, China | 25 community-acquired pneumonia<br><br>20 healthy controls | Bronchoalveolar lavage fluid samples | Samples were collected at bronchoscopic examination in clinical management. | Whole-genome sequencing | The BALF microbiota in COVID-19 patients was similar to that in patients with community-acquired pneumonia. The microbiome had an abundance of oral and upper respiratory commensal bacteria. | No treatment data | None |
| Chen et al. (2020) | 2 patients with COVID-19, China<br><br>A 39-year-old male and a 21-year-old female | None | Bronchoalveolar lavage fluid samples | Samples were collected at bronchoscopic examination for a diagnostic test. | Microbial next-generation sequencing analysis | Authors found *Capnocytophaga* species and *Veillonella* species in BALF samples. | Antiviral and anti-infectious treatment | None |
| Ren et al. (2020) | 5 patients with COVID-19, China<br><br>The median age was 52 (41–65) years<br><br>2 patients were female<br><br>All had moderate to severe COVID-19<br><br>Comorbidities: hypertension, chronic liver disease | None | Bronchoalveolar lavage fluid samples | Samples were collected at bronchoscopic examination during hospitalization. | Microbial next-generation sequencing analysis | Most genome reads were viral (betacoronavirus), with bacterial pathogens such as *Acinetobacter*, *Pseudomonas*, *Chryseobacterium*, *Escherichia*, *Streptococcus*, *Enterococcus*, *Rothia*, and *Lactobacillus*. | 5 patients received antibiotic therapy. Outcomes are recovered (1), hospitalized (3), and died (1). | None |
| Fan et al. (2020) | 20 deceased COVID-19 patients, China<br><br>14 were males<br><br>The median age was 66 years<br><br>Comorbidities: cardiovascular disease (10), hypertension (9), malignancy (7), diabetes (2), chronic kidney disease (2), chronic lung diseases (1) | None | Lung tissue samples | Samples were collected from deceased patients. | Sequencing of V3-V4 regions of bacterial 16S rRNA, ITS gene | *Acinetobacter* was the most common bacterial genus, followed by *Chryseobacterium*, *Burkholderia*, *Brevundimonas*, *Sphingobium*, and *Enterobacterales*. *Cryptococcus* was the most prevalent fungus, along with *Issatchenkia*, *Wallemia*, *Cladosporium*, and *Alternaria*. | All patients received antibacterial and antiviral therapy. | None |
| Tao et al. (2020) | 62 patients with COVID-19, China | 33 seasonal influenza patients<br><br>40 healthy controls | Fecal samples | Samples were collected at first time of visit to the hospital. | Next-generation sequencing of V4 region of the 16S rRNA | COVID-19 patients had an abundance of *Streptococcus*, *Clostridium*, *Lactobacillus*, and *Bifidobacterium* in gut microbiota. In contrast, lower levels of *Bacteroides*, *Roseburia*, *Faecalibacterium*, *Coprococcus*, *Parabacteroides* were found. | No data | Alpha-diversity of gut microbiome decreased in COVID-19 compared with that in healthy control, flu patients. |

*(Continued)*

**Table 1.** (Continued)

| Study | Population and characteristics | Control | Samples or Microbiota community | Timepoint at sample collection | Methods of microbiome characterization | Change in microbiota | Treatment/ Outcome | Other key findings |
|---|---|---|---|---|---|---|---|---|
| Yu et al. (unpublished) | 2 male patients with COVID-19, China<br><br>2 patients with COVID-19, China<br><br>65 years old male and 78 years old male<br><br>Comorbidities: prostatic hyperplasia, chronic bronchitis | 22 healthy cohort (Data imported from Arumugam M, Raes J, Pelletier E, et al. Enterotypes of the human gut microbiome. Nature. 2011;473(7346):174–80.) | Anal swab samples | No data | No data | The proportion of gut microbiota, including *Corynebacterium* and *Ruthenibacterium*, was increased. | Both patients received antiviral and antibacterial agents. and both patients died. | None |
| Ai et al. (unpublished) | 20 patients with COVID-19, China<br><br>10 were female<br><br>The median age was 37 years | 33 pneumonia without COVID-19 | Nasopharyngeal swab samples | Admission day | Multiplex RT-PCR assays<br><br>next-generation sequencing | More than half of patients had co-infection with COVID-19 and another virus, such as influenza A/B, rhino- or enteroviruses, or respiratory syncytial virus. | No data | None |
| Budding et al. (unpublished) | 46 patients with COVID-19, Netherlands | 89 SARS-CoV2 (-) patients | Throat swab samples | Samples were collected for routine diagnostic tests. | 16S rDNA sequencing | *Haemophilus parainfluenzae*, *Neisseria cinerea*,<br><br>*S. mitis*, *S. bovis*, *Leptotrichia buccalis*, and *Rothia mucilaginosa* were the main composition of throat microbiome. | No data | None |
| Xu et al. (a) (abstract only) | No data, China | No data | Intestinal microbiome | No data | No data | COVID-19 patients had intestinal dysbiosis with decreased *Lactobacillus* and *Bifidobacterium*. | No data | None |
| Xu et al. (b) (abstract only) | No data, China | No data | Gut microbiome | No data | No data | Decreased proportions of *Lactobacillus* and *Bifidobacterium* were observed in COVID-19 patients. | No data | None |

depended on each study. Most respiratory samples were collected as part of clinical management, such as diagnostic tests; stool samples were collected at admission, while other studies collected samples during hospitalization.

## Quality of studies

The quality assessment results are summarized in S1 Table. Most studies were generally unclear and of low quality, according to RoBANS.

## Dysbiosis in the fecal microbiome of COVID-19 patients

In five observational studies [11–14, 26], COVID-19 patients had altered intestinal microbiomes compared with the control groups. Three studies by Zuo et al. [11–13] using shotgun metagenomic sequencing of fecal samples described dysbiosis in the bacterial microbiome and mycobiome in COVID-19 patients when compared with those in healthy controls. Notably, COVID-19 patients mainly had increased numbers of opportunistic pathogens (OPs), a part of commensal microbiota that may become pathogenic in the event of host perturbation, such as dysbiosis or impaired immune system (immunocompromised host) [27]. In these reports, the

OPs included *Clostridium hathewayi*, *Actinomyces viscosus*, and *Bacteroides nordii* at the time of hospitalization, and gut bacterial dysbiosis persisted even after SARS-CoV-2 tests using nasopharyngeal swabs or saliva samples became negative. In another report, SARS-CoV-2 RNA was detected in 46.7% of stool samples regardless of the gastrointestinal symptoms [13]. That report also showed that the numbers of specific OPs (e.g., *Collinsella aerofaciens* and *Morganella morganii* spp.) were increased in fecal samples with high SARS-CoV-2 active viral transcription and replication in vitro (infectivity) compared with those in fecal samples with low to no SARS-CoV-2 infectivity [13]. Conversely, fecal samples with low or no SARS-CoV-2 infectivity had increased levels of bacteria belonging to *Parabacteroides*, *Bacteroides*, and *Lachnospiraceae*, which produce short-chain fatty acids (especially butyric acid). Short-chain fatty acids are known to play an important role in boosting the host immunity; thus, these data suggest that OPs pose threats of both reduced host immunity and opportunistic infections in proportion to the load of SARS-CoV-2. In another cohort, reduction in bacterial diversity was described in fecal samples from patients with COVID-19 or influenza compared with that in healthy controls (matched for age, sex, and body mass index) by analyzing the V3–V4 region of the 16S rRNA gene [14]. The study also revealed an increase in the number of OPs such as *Streptococcus*, *Rothia*, *Veillonella*, and *Actinomyces* among COVID-19 patients. Moreover, C-reactive protein, an indicator of bacterial infection, was positively correlated with the bacterial taxa [14].

One gut microbiome study based on 16S rRNA in COVID-19 patients [26] found that the alpha diversity in these patients was lower than that in healthy controls and influenza patients. The abundance of four genera, *Streptococcus*, *Clostridium*, *Lactobacillus*, and *Bifidobacterium*, tended to increase. Conversely, five other genera, *Bacteroides*, *Roseburia*, *Faecalibacterium*, *Coprococcus*, and *Parabacteroides*, showed lower abundance in COVID-19 patients than in control subjects. Furthermore, the study showed that the levels of IL-18, a pro-inflammatory cytokine produced by multiple intestinal cells and the intestinal nervous system, were increased in the sera of COVID-19 patients compared with those in influenza patients and healthy individuals. We also identified one unpublished, non-peer-reviewed article concerning the intestinal microbiome [22]. The study showed that the proportion of pathogenic bacteria in the gut, including *Corynebacterium* and *Ruthenibacterium*, increased in COVID-19 patients compared with another historical cohort data previously reported [22]. In contrast, the abundance of *Bifidobacterium*, *Lactobacillus*, and *Eubacterium* decreased in COVID-19 patients. Moreover, two Chinese articles with English abstracts described dysbiosis with decreased *Lactobacillus* and *Bifidobacterium* levels in some COVID-19 patients [24, 25].

Further, one study reported the fungal microbiome in COVID-19 patients. It showed that the composition of the fecal mycobiome in 30 hospitalized COVID-19 patients was heterogeneous; however, some were enriched in fungal pathogens of *Candida* and *Aspergillus* spp. compared with control subjects. In addition, in some of the COVID-19 patients, intestinal mycobiome instability and prolonged dysbiosis persisted for up to 12 days after the disappearance of SARS-CoV-2 from the nasopharynx [12].

## Severity of COVID-19 and the intestinal microbiome

The association between the severity of COVID-19 and the gut microbiome was mentioned in two studies [11, 14]. The classification of COVID-19 severity in each study are summarized in the S2 File. One study showed the potential importance of *Firmicutes* spp. in the severity of SARS-CoV-2 infection [11]. To understand the association between the gut microbiome and COVID-19 severity, the study assessed the association between the fecal microbiome and COVID-19 severity (mild, moderate, severe) in seven antibiotic-free COVID-19 patients. A

total of 23 bacterial taxa were significantly associated with the severity of COVID-19, and most (15 of 23) were in the phylum *Firmicutes*. Of these, eight classes were positively correlated with disease severity, and seven were negatively correlated. *Firmicutes* spp. were diversely affected in COVID-19 patients. In particular, *Coprobacillus*, *Clostridium ramosum*, and *C. hathewayi* in the *Firmicutes* phylum were the top bacteria showing a positive correlation with the severity of COVID-19.

The other paper reported that COVID-19 patients have lower lymphocyte counts and increased interleukin (e.g., IL-6) and TNF-α levels compared with the healthy cohort [14]. In their study, the differences in gut microbiota abundance, diversity, and structure were not significantly different between patients with mild and severe COVID-19.

## Alterations in the upper respiratory tract microbiome of COVID-19 patients

The upper respiratory microbiome in COVID-19 patients was analyzed in two observational studies (one using nasopharyngeal swabs and the other using nasopharyngeal swabs and sputum) [15, 16]. One study found that the nasopharyngeal microbiome was not significantly different between 18 acute-phase COVID-19 patients and 12 control individuals (SARS-CoV-2 negative) upon analyzing the V5–V6 regions of the 16S rRNA gene in the nasopharyngeal swab samples [15]. The other study performed a metatranscriptomic analysis in a cohort of 113 patients (38 patients with COVID-19 and 74 patients with non-COVID-19 pneumonia). The Shannon index of nasopharyngeal swab specimens showed little difference between both groups. In contrast, the Shannon index of sputum specimens was significantly lower in COVID-19 patients, suggesting that they may have an altered middle pharynx microbiome compared with that in the other patients with non-COVID-19 pneumonia. In that report, co-infection analysis also identified 24 potentially pathogenic microorganisms in 18 of the 38 (47.4%) COVID-19 patients. Among them, 16 different microorganisms were detected, with the OPs *Candida albicans* and *human alpha-herpesvirus* 1 being the most abundant. Moreover, the remaining eight microorganisms were viral pathogens, including the human influenza virus and respiratory syncytial virus [16]. We identified two unpublished (non-peer-reviewed) articles examining the upper respiratory tract microbiome [21, 23]. Using next-generation sequencing of nasopharyngeal swabs, one study found that 11 of 20 patients had co-infection with COVID-19 and another virus, including influenza A/B, rhino- or enteroviruses, or respiratory syncytial virus [21]. The other study described the main microbiome composition of COVID-19 throat samples to include *Haemophilus parainfluenzae*, *Neisseria cinerea*, *Streptococcus mitis*, *Streptococcus bovis*, *Leptotrichia buccalis*, and *Rothia mucilaginosa*. The authors also observed that the pharyngeal microbiome diversity of all examined bacterial phyla (e.g., *Bacteroidetes*, *Proteobacteria*) decreased in older adults than younger adults, both having COVID-19, which could explain the differences in severity [23].

## Alterations in the lower respiratory tract microbiome of COVID-19 patients

One study on whole-genome sequencing of BALF samples of COVID-19 patients, community-acquired pneumonia (CAP) patients, and healthy controls found that the microbiota in COVID-19 patients was similar to that in CAP patients. However, the microbiomes of both the COVID-19 and CAP groups were significantly different from those of healthy controls, who demonstrated enrichment with known oral and upper respiratory commensal bacteria, suggesting dysbiosis in the lung biota of COVID-19 patients [17]. Another study based on 16S rRNA sequences found that *Acinetobacter* was the most common bacterial genus, followed by

*Chryseobacterium*, *Burkholderia*, *Brevundimonas*, *Sphingobium*, and *Enterobacterales* in lung tissues of deceased COVID-19 patients [18]. Moreover, the study identified *Cryptococcus* as a prevalent fungus, along with *Issatchenkia*, *Wallemia*, *Cladosporium*, and *Alternaria*. A case study using whole-genome sequencing reported co-infection of *Capnocytophaga* and *Veillonella* species in the BALF samples of one out of two COVID-19 patients, but the pathogenicity was unknown [19]. Another case series based on whole-genome sequencing of samples from five COVID-19 patients found that most non-human genomic reads from BALF samples were from a beta coronavirus genome. A few bacterial genomes, including those of *Acinetobacter*, *Pseudomonas*, *Chryseobacterium*, *Escherichia*, *Streptococcus*, *Enterococcus*, *Rothia*, and *Lactobacillus*, were also detected [20].

## Discussion

Our systematic review accumulated the evidence in the intestinal and respiratory microbiome in hospitalized COVID-19 patients. Several studies indicated the fecal, respiratory microbiome dysbiosis with increasing OPs [11–13, 16, 17]. One observational study showed that COVID-19 severity could be associated with the baseline fecal microbiome [11]. However, the extent to which the corresponding human microbiome contributes to COVID-19 remains unclear. Further microbiome research, including longitudinal data (e.g., post-hospitalization), studies from various countries and ethnicities, and evidence of severity is needed.

The common intestinal microbiota comprises four domains, including *Firmicutes* (i.e., *Clostridium*, *Bacillus*, and *Lactobacillus*), *Bacteroides*, *Actinobacteria* (i.e., *Bifidobacterium*), and *Proteobacteria* [27, 28]. The dysbiosis of the intestinal flora is currently considered to be associated with the host antiviral immune response. Previous *in vivo* and *in vitro* studies have revealed that the intestinal microbiome could affect innate and adaptive immune responses to viral infections such as influenza [29–33]. Results of two independent studies revealed an increase in OPs in the gut of COVID-19 patients with prolonged dysbiosis [11, 14]. Similarly, we showed that about 20 Japanese COVID-19 patients also had dysbiosis associated with increased OPs compared with healthy cohorts by 16S rRNA analysis (unpublished data). Although the findings suggested a relationship between OPs and the intestinal microbiome, the correct understanding of the enrichment and pathogenicity of OPs remains uncertain. The role of these OPs may be connected to potential secondary bacterial infection when the host immune system is impaired [11]. Further research is needed to address the mechanism of harm by these OPs better.

Disruption of the intestinal microbiome is also reported to be a predisposing factor for pro-inflammatory conditions (e.g., sepsis) [34], although the mechanism of intestinal microbiome change underlying severe conditions is uncertain. A previous study related to inflammation showed that patients with systemic inflammatory response syndrome had a low abundance of gut anaerobes such as *Bifidobacterium* and *Lactobacillus* and high OP abundance [35]. Similarly, the gut microbiome dysbiosis with an abundance of OPs occurred in patients with COVID-19 [11–14, 26]. One study showed that *Faecalibacterium* was negatively correlated with severe COVID-19 [11], although *Faecalibacterium* has been reported to be associated with a pathway to reduce intestinal inflammation [34]. In this review of the microbiome in COVID-19, few studies reported intestinal microbiota alteration and severe COVID-19. Additional comparative studies on COVID-19 severity, including the convalescent phase of COVID-19 and the phase of antibody production against SARS-CoV-2, are needed. The dysbiosis of human microbiotas could be one possible host factor underlying severe COVID-19.

Microbiomes of the upper and lower respiratory tracts have been reported to have similar microbiological characteristics [28]. The upper respiratory microbiome influences respiratory

viral infection, including disease severity [36], vaccine response, and secondary bacterial infection [37]. Similarly, the lower respiratory microbiome, including the lung microbiome, shows decreased diversity in ARDS patients without COVID-19, with increased abundance of *Staphylococcus*, *Streptococcus*, and *Enterobacterales* species and increased IL-6 levels [38]. Our review showed that the airway microbiome of COVID-19 patients was different from that of healthy controls. There was no difference between the microbiomes of COVID-19 and CAP patients, but this observation may be attributed to the use of antibiotics for these patients, affecting the composition of their microflora in the same way. In contrast, the nasopharyngeal microbiome may be less affected by acute respiratory virus infection. A previous study revealed that the nasopharyngeal microbiome was not different between patients with influenza and control individuals [39]. Two independent studies [15, 16] on nasopharyngeal swabs showed no significant difference in the microbiomes of COVID-19 patients and control subjects. In the future, more data on the upper and lower airway microbiomes would provide a detailed understanding of the relationship between respiratory virus infection, including COVID-19, and the microbiome.

Currently, little is known about the relationship between human microbiota and COVID-19. The review revealed several necessary points that warrant future research on the microbiome in COVID-19. First, longitudinal data, including post-hospitalization information, are lacking. Besides the poor understanding of the pathogenesis of COVID-19, there is limited knowledge about the clinical sequelae that may persist after viral clearance because COVID-19 is a new disease. It is important to establish a longitudinal cohort that has recovered from COVID-19 and investigate the association between human microbiota and clinical features of acute infection of the respiratory tract. Second, reports from China dominate this review. Data from other countries are needed, as ethnicity is known to be a significant factor in the diversity of the microbiome. Our results have a population bias and may not be generalizable. Third, analysis of the human microbiome of non-intestinal organs, such as the respiratory tract, oral cavity, skin, or urinary tract, will prove fruitful in understanding the microbiota in COVID-19 patients. Fourth, additional microbiome data regarding their correlation to COVID-19 severity are essential. If the gut microbiota profile for each disease severity becomes clear in the future. The microbiome may become a prognostic marker for disease progression. Furthermore, in that case, early control of the gut microbiota (e.g., by symbiotics, probiotics, fermented food, and fecal transplant) would be effective in the aspect of prevention or therapeutics. Several trials, including randomized controlled trials for the efficacy of probiotics for COVID-19 (Cochrane Central Register of Controlled Trials: ChiCTR2000029974, NCT04390477), are ongoing [40]. To date, the effects of altering the gut microbiome remain uncertain, and the results of the ongoing trials are needed.

This review has several limitations. First, only English publications were searched; literature in other languages, including Chinese, was not included. Second, this review did not consider antibiotic and probiotic use. Not all studies documented antibiotic and probiotic use, which could be a potential factor affecting the microbiome findings. Third, various approaches, such as whole-genome sequencing and 16S rRNA sequencing, were used to assess the microbiota, and the spectrum of detected microorganisms (bacteria, fungi, viruses) differed for each study. Fourth, the causality between the microbiome and disorder is uncertain. The question of which came first, the chicken or the egg, is a classic problem of the microbiome. Therefore, longitudinal data on the microbiome in COVID-19 patients are needed. Fifth, our review lacked metabolomics studies. Although it is unclear whether changes in the bacterial flora are a cause or consequence, changes in secondary metabolites secreted by bacteria may lead to changes in the gut environment, which may be related to the pathogenesis of the disease by

leading to some inflammatory stimuli. Further elucidation of the correlation between metabolomics and human microbiome due to SARS-CoV-2 infection is expected.

In conclusion, limited evidence implies that COVID-19 patients had altered gut and respiratory tract microbiomes along with an increased abundance of OPs.

## Supporting information

**S1 File. The results of database search.**
(DOCX)

**S2 File. The classification criteria of COVID-19 severity.**
(DOCX)

**S1 Table. Risk of bias for each study.**
(DOCX)

**S1 Checklist. PRISMA_2009_checklist.**
(DOC)

## Author Contributions

**Conceptualization:** Shinya Yamamoto.

**Data curation:** Shinya Yamamoto, Azumi Tamura.

**Formal analysis:** Shinya Yamamoto, Azumi Tamura.

**Investigation:** Shinya Yamamoto, Azumi Tamura.

**Methodology:** Shinya Yamamoto, Makoto Saito.

**Resources:** Shinya Yamamoto.

**Supervision:** Shinya Yamamoto, Makoto Saito, Taketoshi Mizutani.

**Visualization:** Shinya Yamamoto, Azumi Tamura.

**Writing – original draft:** Shinya Yamamoto.

**Writing – review & editing:** Makoto Saito, Azumi Tamura, Diki Prawisuda, Taketoshi Mizutani, Hiroshi Yotsuyanagi.

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
