## [Decision Letter · Decision Letter 0]

12 Apr 2021

PONE-D-21-06559

The human microbiome and COVID-19: A systematic review

PLOS ONE

Dear Dr. Yamamoto,

Thank you for submitting your manuscript to PLOS ONE. After careful consideration, we feel that it has merit but does not fully meet PLOS ONE’s publication criteria as it currently stands. Therefore, we invite you to submit a revised version of the manuscript that addresses all the points raised during the review process by the 3 reviewers.

Particularly, you will see that all reviewers although finding your work of great interest, asked for additional information regarding notably the precise methodology used for structuring the review, the criteria of severity used in the different studies cited, the control of possible bias and inherent limitation regarding the individual studies referenced. 

We look forward to receiving your revised manuscript.

Kind regards,

Francois Blachier, PhD

Academic Editor

PLOS ONE

Journal Requirements:

3, Please include captions for your Supporting Information files at the end of your manuscript, and update any in-text citations to match accordingly. Please see our Supporting Information guidelines for more information: http://journals.plos.org/plosone/s/supporting-information.

Reviewers' comments:

Reviewer's Responses to Questions

**Comments to the Author**

1. Is the manuscript technically sound, and do the data support the conclusions?

Reviewer #1: Partly

Reviewer #2: Yes

Reviewer #3: Yes

2. Has the statistical analysis been performed appropriately and rigorously? 

Reviewer #1: No

Reviewer #2: Yes

Reviewer #3: N/A

3. Have the authors made all data underlying the findings in their manuscript fully available?

Reviewer #1: Yes

Reviewer #2: Yes

Reviewer #3: Yes

4. Is the manuscript presented in an intelligible fashion and written in standard English?

Reviewer #1: Yes

Reviewer #2: Yes

Reviewer #3: Yes

5. Review Comments to the Author

Reviewer #1: Even if it seems interesting to study the potential association between the dysbiosis and the immune response to the SARS_CoV_2, it should have been done with more well defined methodology. What is the real scope of the review ? Association between severity of the disease and ? With which criterias ? How the biais are controlled in the studies selected ?

The methodology of the studies retained are not well enough detailed. So it is impossible for the reader to assess the validity of the association described.

Reviewer #2: Yamamoto et al. conducted a systematic literature review on the association between the

microbiome and COVID-19 in humans. They shown that human fecal and respiratory tract microbiome changed in COVID-19 patients with opportunistic pathogen abundance. The quality of the process to conducted this study is excellent. Please find my comments.

- The conclusion need to be more shorter with a highlight in terms of Microbiome dysbiosis.

- Do you think that there is non supplementation with probiotics or prebiotics or fermented food in patients ?

- Can you added your recommendations or can you speculate if patients can be supplemented with probiotics ?

- Can microbiota dysbiosis be uses as a signature of COVID-19 severity (as a biomarker)?

- The aspect of microbiota or microbiome modulation in patients can be discussed.

- The prevention or therapeutic aspects : can you comments please?

- One of the limitation may be the lack of metabolomics studies, can you comments on this point.

Reviewer #3: In this systematic review, the authors collected evidence on alterations in the gut, upper and lower respiratory microbiota in Covid-19 infection.

The study is well conducted and written, offering a good perspective on the work in literature.

The evidence collected is unfortunately of low grade. All the studies collected had an observational design, 14/16 are conducted in China and consider only hospitalized patients. For these reasons, many confounding factors may have altered the results. Moreover, different types of samples and different methods of analysis are used.

Major revision

- Clarify whether unpublished studies were reviewed by a third-party researcher not involved in the study.

Minor revision

- The design and characteristics of the included studies are well described in Table 1. However, they could be reported more clearly in the text.

- In the "Selection criteria and search strategy" paragraph, words used for the research could be placed in quotation marks rather than in parentheses.

6. PLOS authors have the option to publish the peer review history of their article (what does this mean?). If published, this will include your full peer review and any attached files.

Reviewer #1: No

Reviewer #2: **Yes: **Marvin Edeas, MD, PhD

Professor of Medicine

Université de Paris, INSERM U1016

Institut Cochin

Department Endocrinology, Metabolism

and Diabetes

Faculté de médecine Cochin-Port Royal

24 rue du Faubourg St Jacques,

75014 Paris-France

Reviewer #3: No

---

## [Author Response · Author response to Decision Letter 0]

17 May 2021

Reviewer #1: Even if it seems interesting to study the potential association between the dysbiosis and the immune response to the SARS_CoV_2, it should have been done with more well defined methodology. What is the real scope of the review ? Association between severity of the disease and ? With which criterias ? How the biais are controlled in the studies selected ?

The methodology of the studies retained are not well enough detailed. So it is impossible for the reader to assess the validity of the association described.

Response: Thank you for your comment. All of your suggestions have been incorporated into our revised manuscript. The scope of review has been provided in the Introduction section (Page 4, line 72−74)” Moreover, in this review, the relationship between the human microbiome and COVID-19 severity has been investigated”. The severity criteria have been added to the Result (Page 8,9 line 189-190) and Supplementary sections. We assessed the risk of bias in each study (Supplementary Table 1).

Reviewer #2: Yamamoto et al. conducted a systematic literature review on the association between the microbiome and COVID-19 in humans. They shown that human fecal and respiratory tract microbiome changed in COVID-19 patients with opportunistic pathogen abundance. The quality of the process to conducted this study is excellent. Please find my comments.

- The conclusion need to be more shorter with a highlight in terms of Microbiome dysbiosis.

Response: Thank you for your comment. The conclusion section has been shortened as suggested. Page 15. line 354−355. “In conclusion, limited evidence implies that COVID-19 patients had altered gut and respiratory tract microbiomes along with an increased abundance of OPs.”

- Do you think that there is non supplementation with probiotics or prebiotics or fermented food in patients ?

- Can you added your recommendations or can you speculate if patients can be supplemented with probiotics ?

- Can microbiota dysbiosis be uses as a signature of COVID-19 severity (as a biomarker)?

- The aspect of microbiota or microbiome modulation in patients can be discussed.

- The prevention or therapeutic aspects : can you comments please?

Response: Thank you for your comment. All of your suggestions have been incorporated into our revised manuscript. Information on probiotics, prebiotics, or fermented food and microbiome modulation and clinical use of microbiome have been provided in the discussion section. Page 14, line 329-332. “If the gut microbiota profile for each disease severity becomes clear in the future. The microbiome may become a prognostic marker for disease progression. Furthermore, in that case, early control of the gut microbiota (e.g., by symbiotics, probiotics, fermented food, and fecal transplant) would be effective in the aspect of prevention or therapeutics..”

- One of the limitation may be the lack of metabolomics studies, can you comments on this point.

Response: Thank you. A relevant text has been added to the limitation paragraph of the Discussion section. Page 15, line 347−352. “Fifth, our review lacked metabolomics studies. Although it is unclear whether changes in the bacterial flora are a cause or consequence, changes in secondary metabolites secreted by bacteria may lead to changes in the gut environment, which may be related to the pathogenesis of the disease by leading to some inflammatory stimuli. Further elucidation of the correlation between metabolomics and human microbiome due to SARS-CoV-2 infection is expected..”

Reviewer #3: In this systematic review, the authors collected evidence on alterations in the gut, upper and lower respiratory microbiota in Covid-19 infection.

The study is well conducted and written, offering a good perspective on the work in literature.

The evidence collected is unfortunately of low grade. All the studies collected had an observational design, 14/16 are conducted in China and consider only hospitalized patients. For these reasons, many confounding factors may have altered the results. Moreover, different types of samples and different methods of analysis are used.

Major revision

- Clarify whether unpublished studies were reviewed by a third-party researcher not involved in the study.

Response: Thank you for your suggestions. The unpublished studies (Yu et al. 2020, Budding AE, et al. 2020, Ai JW, et al.2020) reported were not peer-reviewed by a third-party researcher. This information has been provided in the revised manuscript. Page 5, line 114.

Minor revision

- The design and characteristics of the included studies are well described in Table 1. However, they could be reported more clearly in the text.

Response: Thank you for your comment. Revisions describing the included studies have been made to the revised manuscript (Page 7, line 143−144, 145, 148−149, 157,160-161. Page 8, line 174-175; Page 9, line 196, 212; Page 11, line 244-245).

- In the "Selection criteria and search strategy" paragraph, words used for the research could be placed in quotation marks rather than in parentheses

Response: Thank you for your suggestion. Each searched terms were placed in quotation marks as suggested, and some round brackets that are needed as an operator for the search condition were retained.

---

## [Decision Letter · Decision Letter 1]

2 Jun 2021

The human microbiome and COVID-19: A systematic review

PONE-D-21-06559R1

Dear Dr. Mizutani,

We’re pleased to inform you that your manuscript has been judged scientifically suitable for publication and will be formally accepted for publication once it meets all outstanding technical requirements.

Kind regards,

Francois Blachier, PhD

Academic Editor

PLOS ONE

Additional Editor Comments (optional):

Reviewers' comments:

Reviewer's Responses to Questions

**Comments to the Author**

1. If the authors have adequately addressed your comments raised in a previous round of review and you feel that this manuscript is now acceptable for publication, you may indicate that here to bypass the “Comments to the Author” section, enter your conflict of interest statement in the “Confidential to Editor” section, and submit your "Accept" recommendation.

Reviewer #1: All comments have been addressed

Reviewer #2: All comments have been addressed

2. Is the manuscript technically sound, and do the data support the conclusions?

Reviewer #1: Yes

Reviewer #2: Yes

3. Has the statistical analysis been performed appropriately and rigorously? 

Reviewer #1: N/A

Reviewer #2: Yes

4. Have the authors made all data underlying the findings in their manuscript fully available?

Reviewer #1: Yes

Reviewer #2: Yes

5. Is the manuscript presented in an intelligible fashion and written in standard English?

Reviewer #1: Yes

Reviewer #2: Yes

6. Review Comments to the Author

Reviewer #1: The authors are very clear with the fact that the article is exploratory. It is still quite difficult to extrapolate the observations reported in the review and more, it is not simple to imagine any clinical implications of these observations. Nevertheless, it is interesting to observe that, like for other diseases, dysbiosis is associated with covid19. All the results of the articles included in the review are well reported.

Reviewer #2: Thanks for all your comments and the clarifications. The paper is excellent and will provide a strong contribution to this field.

7. PLOS authors have the option to publish the peer review history of their article (what does this mean?). If published, this will include your full peer review and any attached files.

Reviewer #1: No

Reviewer #2: **Yes: **Marvin EDEAS

---

## [Editor Report · Acceptance letter]

14 Jun 2021

PONE-D-21-06559R1 

The human microbiome and COVID-19: A systematic review 

Dear Dr. Mizutani:

I'm pleased to inform you that your manuscript has been deemed suitable for publication in PLOS ONE. Congratulations! Your manuscript is now with our production department. 

Kind regards, 

on behalf of

Dr. Francois Blachier 

Academic Editor

PLOS ONE